# OpenReview forum: "Jailbreaking Leading Safety-Aligned LLMs with Simple Adaptive Attacks"
_ICLR.cc/2025/Conference — ICLR 2025 Poster_

### Official Review · Reviewer_QTMV · 2024-10-31

**Soundness:** 3
**Presentation:** 2
**Contribution:** 3
**Rating:** 6
**Confidence:** 5

**Summary:**

This paper uses the adaptive method based on logprob to jailbreak well-aligned LLMs, and achieves the near-perfect success rates.

**Strengths:**

+ The method is straightforward and easy to understand
+ The attack success ratio is impressive with such straightforward methods
+ The evaluation is comprehensive, with many well-aligned models evaluated


First of all, I think the method of the paper is very straightforward. However, with such a straightforward method, it could achieve a near-perfect attack success rate in the black-box setting. It shows the signal of attack saturation even in the black-box setting, which needs us to do more study on the defense side instead of on the attack side.

To be honest, I have seen and reviewed many jailbreak attack papers that are either too complicated with the agentic design or too simple that can be reduced to a new prompting trick, while most of them just achieve marginal improvement compared with old jailbreak papers such as PAIR. With those papers flooding into ML, AI, security and NLP domain, it not only degrades the quality of the conferences but also wastes the reviewer's efforts and makes it much more challenging to find a really impactful LLM alignment work. Since this work indicates the saturation of the attack success ratio, then it shows that the current defense or alignment is not comprehensive and the LLM security community should focus more on better alignment. Also, for those works with just some prompting diffs that could be somewhat effective and lead to higher (or not) attack success rate compared with old works, I wish this paper could act as the whistleblower. If the jailbreak attack success ratio is not even near-perfect, then we should let the community know it is better to reconsider whether to submit such work (or even consider whether to do such work before operation). I wish we could set a higher bar for the LLM security community, especially for the jailbreak attack works.

**Weaknesses:**

- Lack of further discussion of the insights of the paper
- Lack of baseline
- Lack of potential defense
- Lack of ethical consideration and response disclosure

While I appreciate this work, I also have higher expectations for this work in our community, and thus I have identified some weaknesses that could make this work better. First, as discussed in the strengths, I think the discussion of this paper could go further to give better insights to the community. Second, I understand that for some models there are no established baselines and some baseline methods show significantly lower attack success rates, as a research paper, I still recommend having some well-known baselines reproduced instead of just directly referring to the reported numbers in prior works. It would make the evaluation more convincing. Also, I would suggest testing the attack with some defense approaches. I am curious how the attack would perform when there are defenses like SmoothLLM and RPO. Lastly, since it is an attack paper and it shows strong attack capability, I think there's a need for ethical concern discussion and what's your efforts to mitigate the ethical concern, such as response disclosure to the LLM developer you tested in your work.

Some minor issues: some models' versions are not clear enough, like what's the specific version of gpt4o and llama-3.

**Questions:**

See the weakness above, I **can raise the score** if the authors could address all these concerns

**Details Of Ethics Concerns:**

This is a jailbreak attack and shows strong attack performance, which could be used with malicious intent.

---

> ### Comment · Reviewer_QTMV · 2024-11-23
>
> Due to the inactivity of author response and looking at other reviewer’s comment, I feel reserved for the contribution of the work, and has decrease my score to 6.

---

> ### Author Response · Authors · 2024-11-24
> **Rebuttal**
>
> We thank the reviewer for the positive comments and score. Please note that **we received seven reviews**, most of which including multiple detailed questions. Thus, we took advantage of the entire rebuttal time to provide thorough responses as well as new experiments. Moreover, the discussion phase **still has 3 full days** until November 26, AoE (Tuesday), and we are ready to discuss follow-up questions.
>
> ---
>
> A key goal of our paper was indeed to show how simple, yet adaptive, black-box methods can achieve near-perfect jailbreaking success rate against leading LLMs. Moreover, we provide actionable approaches and recommendations to test future models. While the end-goal is to find a single approach effective across LLMs, this objective currently seems out of reach, and is likely to become even more challenging with the introduction of specialized defenses. In the following we address the specific concerns raised in the review.
>
>
> **Lack of further discussion of the insights of the paper.**
> The main insights provided by our work are summarized in the last paragraph of the introduction and in Section 6, which includes paragraphs on **Our findings** and **Recommendations**. Given the space constraints in the main part, we could not expand further on this, although we believe we summarized our main points there.
>
> **Lack of baseline.**
> We agree that the comparison to prior work could have been more direct. To address this concern, here we present a direct comparison with [GCG](https://arxiv.org/abs/2307.15043), [PAIR](https://arxiv.org/abs/2310.08419), and a popular template attack (“Always Intelligent and Machiavellian” (AIM) from http://jailbreakchat.com/) on 100 JBB-Behaviors from JailbreakBench using a different semantic judge (Llama-3 70B instead of GPT-4 with a different prompt). The numbers below are attack success rates for four different models:
>
> | Attack           | Vicuna 13B | Llama-2-Chat-7B | GPT-3.5 | GPT-4 Turbo |
> |-------------------|--------|---------|---------|-------|
> | GCG              | 80%    | 3%      | 47%     | 4%    |
> | PAIR             | 69%    | 0%      | 71%     | 34%   |
> | Jailbreak-Chat          | 90%    | 0%       | 0%      | 0%    |
> | Prompt Template + Random Search + Self-transfer  | 89%    | 90%     | 93%     | 78%   |
>
> We would like to highlight the particularly large gap on Llama-2-Chat-7B: 90% ASR with our method vs. 3% ASR of plain GCG. Also, we note that the Llama-3 70B judge is in general *stricter* than the GPT-4 judge used as a stopping criterion for the attack, which explains a general drop of ASRs from 100% to the 80%-90% range. However, we expect that with more random restarts and by using the target judge model as a stopping criterion, the ASR of our attacks will go up to 100%. We will add and discuss these results in the paper.
>
> **Lack of potential defense.**
> We agree with the reviewer that developing adaptive jailbreaks against defended models would be the next logical step. However, the main goal of our work was to demonstrate that currently deployed LLMs, while very capable, are vulnerable to simple jailbreaking attacks. We anticipate that bypassing defenses will require new adaptive algorithms in at least some cases, which could constitute non-trivial standalone projects.
>
> **Lack of ethical consideration and response disclosure.**
> We want to remark that we disclosed our findings about the vulnerability of Claude models to the pre-filling attacks with Anthropic significantly before the submission of the manuscript (in March 2024). Apparently, this vulnerability was not considered severe enough to be quickly fixed. Our understanding is that the main LLM companies follow some version of a responsible scaling policy (https://www.anthropic.com/news/anthropics-responsible-scaling-policy) that suggests that the current models (ranked at AI Safety Level 2) do not pose substantial public harm. Thus, fixing existing jailbreaks, such as those explored in our work, is perhaps not a high priority for them. We will add a paragraph discussing potential ethical concerns about our work in the appendix.
>
> **Some minor issues: some models' versions are not clear enough, like what's the specific version of gpt4o and llama-3.**
> We mentioned the version of GPT-4o in Line 356 (i.e., we used `gpt-4o-2024-05-13`). For Llama-3 models, we used the original Llama-3 checkpoints (i.e., not Llama-3.1). We will clarify these details in the revised version and make them more prominent.
>
> > See the weakness above, I can raise the score if the authors could address all these concerns
>
> We thank you again for reading our paper in detail. We hope our rebuttal addresses your concerns.

---

> > ### Comment · Reviewer_QTMV · 2024-11-25
> >
> > Thank you for the detailed rebuttal. However, after reviewing your responses, I find that several of my major concerns remain unresolved. Given that I have already provided a positive score, I will maintain my current rating. Below, I elaborate on the reasons for my decision:
> >
> > 1.  **Revisions and Promises**: While ICLR allows for revisions, the proposed changes and additional experiments are not currently reflected in the submitted manuscript. It is challenging to assess the quality and impact of these promised revisions solely based on your responses. A formal revised paper with these additions would provide a clearer picture for evaluation.
> >
> > 2.  **Depth of Discussion**: The current discussion in the paper remains superficial. While you highlight the success of your approach, deeper insights into why such a straightforward method achieves near-perfect ASR across well-aligned models are missing. Key questions remain unaddressed: Do we still need white-box optimization or agentic jailbreak methods? How can LLMs be better defended against your adaptive approach? These questions are critical for advancing the field and should be explored in greater depth.
> >
> > 3.  **Baseline Comparisons**: While I appreciate your inclusion of comparisons with GCG and PAIR, I believe that additional state-of-the-art baselines should be included, such as GCG++ (white-box) and "GPT-4 Jailbreaks Itself with Near-Perfect Success Using Self-Explanation" (black-box). Moreover, the experimental setup for the baselines you report is unclear. For instance, did you use transfer GCG or directly attack open-sourced models like Llama-2 with white-box access? What system prompts were used? It is also surprising that your results for GCG on Llama-2-7B-Chat are significantly lower than reports in original paper, other works, as well as my personal experience. I can have around 20% ASR for Llama-2-7B-Chat using the official system prompt: "You are a helpful, respectful, and honest assistant. Always answer as helpfully..." Clarifying these discrepancies is essential.
> >
> > 4.  **Evaluation Against Defenses**: While I understand this is primarily an attack paper, testing your method against existing defense mechanisms such as noise-based approaches (e.g., PRO, SmoothLLM) or prompt-based defenses (e.g., sandwich prompt, self-reminder) would provide valuable insights. Even if the ASR drops significantly, this would still offer important contributions to understanding the robustness of your method. Conversely, if the ASR remains high, it would demonstrate the limitations of current defenses. Either outcome would enrich the paper.
> >
> > 5.  **Ethical Concerns and Broader Impact**: I do not agree with the claim that current safety alignment is sufficient, as academic research often aims to achieve higher alignment goals beyond what is profitable for companies. Given the high ASR of your proposed attack and its potential for malicious use, the ethical considerations and broader impact should be more explicitly reflected in your paper. This is particularly important given the potential harm posed by such effective jailbreak techniques.
> >
> >
> > In summary, while I recognize the merits of your work, the issues outlined above remain unresolved. I **strongly recommend addressing these concerns in a formal revision**, as this would help the community better evaluate and understand the contributions of your work. For now, I choose to maintain my score.

---

> > > ### Author Response · Authors · 2024-11-25
> > > **Thanks for the follow-up discussion**
> > >
> > > Thanks for the follow-up discussion!
> > >
> > > We note that having a wide range of baselines *is*, of course, important in general. However, it is not particularly important *for our work*, since we have achieved 100% success rate on all models (with the exception of 96% on GPT-4 Turbo). Any other baseline would have $\leq$ 100% success rate, and thus will not change our main claims.
> > >
> > > As for the ethical concerns, it is not a question of profitability, but a question of the amount of harm coming from the current LLMs. Currently, there exist models without any guardrails (like Mistral) and their misuse has been minimal. Nonetheless, we did report our results to Anthropic in March 2024 to make sure they have a chance to fix the prefilling option, if they are willing to. We are definitely happy to include all these considerations in the next version of the paper.
> > >
> > > As for defenses, **we have evaluated 21 models** in our paper (see Table 21), including many defenses implemented via supervised fine-tuning, RLHF, and other alignment methods. Moreover, all Llama models were evaluated with a prompt defense (i.e., the safety prompt from the Llama-2 paper showed in Table 9). Respectfully, asking us to evaluate *even more* models/defenses would be a little bit too much :-)

---

> > > > ### Author Response · Authors · 2024-11-27
> > > > **Paper revision**
> > > >
> > > > We thank again the reviewer for the detailed comments.
> > > >
> > > > > I **strongly recommend addressing these concerns in a formal revision**, as this would help the community better evaluate and understand the contributions of your work.
> > > >
> > > > We have incorporated the points we have been discussing in the revised version of the paper. It did take us some time due to a large number of reviews (i.e., seven) and suggestions. We hope the new version addresses your concerns much more concretely.
> > > >
> > > > We are happy to provide more input if needed!

---

### Official Review · Reviewer_2GYX · 2024-11-01

**Soundness:** 3
**Presentation:** 3
**Contribution:** 2
**Rating:** 5
**Confidence:** 4

**Summary:**

This paper proposes an adaptive attack that specifies a set of prompt templates and uses a random search algorithm to iteratively craft jailbreaking prompts based on the templates till it leads to successful jailbreaking. The paper also tests the transferability of the attack and a direct prefilling attack. The paper evaluates a large number of models in the evaluation section.

**Strengths:**

+ The evaluation section covers a large number of models.

+ The paper summarizes key findings and provides recommendations for better model robustness.

**Weaknesses:**

- I have concerns regarding the technical novelty of the proposed technique. In addition, the proposed method's effectiveness relies on the proposed adversarial prompt templates, where the authors do not provide a scientific way of obtaining them. Overall, the paper is more like a technical report with a set of methods that the authors find useful after a large number of trials. It lacks the rigorous scientific designs, challenges, and solutions that are required and valued in a research paper.

- Although discussing them, the paper does not compare the proposed method with the key-related work. It again is a concern for a research paper. To demonstrate that the proposed method is effective, IMOH, I think it is necessary to compare it with existing works.

- The evaluation setup is not clear. For example, it is not clear what is the tasks and data for jailbreaking each model. Not sure if the goal of attacking each model is to force the model to answer harmful questions or output toxic or biased outputs regardless of input. It should affect the judgment model as well, which is also not specified in the paper. The setup of the trojan attack is clear, not sure if the attack above shares the same setups and goals as the trojan attacks.

- The proposed attack is still heavily emphasized on appending prefix or suffix tokens, which is not diverse or realistic enough.

**Questions:**

1. What are the key technical challenges and contributions of the proposed method?

2. How to come up with the adversarial prompt templates? How much the proposed method will affect in performance of the template changes?

3. Can we improve the proposed attack by replacing random search with some more effective search, such as fuzzing or RL [1]?

4. What are the setups of the jailbreaking attacks and what are the judgment models?

5. How to justify that the proposed method can test the model comprehensively and provide realistic attack prompts?

[1] When LLM Meets DRL: Advancing Jailbreaking Efficiency via DRL-guided Search

---

> ### Author Response · Authors · 2024-11-24
> **Rebuttal (Part 1)**
>
> We thank the reviewer for the detailed feedback.
>
> > I have concerns regarding the technical novelty of the proposed technique. In addition, the proposed method's effectiveness relies on the proposed adversarial prompt templates, where the authors do not provide a scientific way of obtaining them. Overall, the paper is more like a technical report with a set of methods that the authors find useful after a large number of trials. It lacks the rigorous scientific designs, challenges, and solutions that are required and valued in a research paper.
>
> From a technical standpoint, we propose novel approaches for crafting jailbreaks attacks: 1) we show how to leverage random search for attacking LLMs which expose top-k logprobs, 2) we propose in-context jailbreaking and self-transfer, two novel techniques which contribute to successfully jailbreak GPT-4o, R2D2, and the Llama models, 3) we uncover how the pre-filling option in Claude models, which otherwise provide very restricted access, can be exploited for jailbreaks.
>
> Moreover, while our main prompt template has been obtained via manual search, it has been employed on multiple target LLMs, with only small adjustments for GPT-4o. Moreover, recent work has used our prompt template for jailbreaking LLM-based agents. This shows that it can serve as a starting point for jailbreaking future models, to be refined on a case-by-case basis.
>
> Finally, we disagree with the reviewer on the nature of our work: we tackle a very well-defined problem (jailbreaking aligned LLMs) studied by lots of prior works, we propose a set of methods, report their results, and compare them with existing works. Our attacks achieve 100% success rate on safety-aligned LLMs for the first time, which shows the relevance of our works for the community.
>
> ---
>
> > Although discussing them, the paper does not compare the proposed method with the key-related work. It again is a concern for a research paper. To demonstrate that the proposed method is effective, IMOH, I think it is necessary to compare it with existing works.
>
> We remark that we already include several baselines (when available) for each target LLM (see complete results in Table 21). Given the fast pace of the field and the cost of running some of the baselines, it is not feasible to compute the results of every existing method on every target model (some of these models have even appeared way after the attacks have been proposed). We are nevertheless happy to add new baseline results, such as many-shot jailbreaking and Pliny’s attacks (as suggested by **Reviewer 82AN**). However, we highlight that our attacks achieve a 100% success rate, thus would outperform or match any other baseline.
>
> ---
>
> > The evaluation setup is not clear. For example, it is not clear what is the tasks and data for jailbreaking each model. Not sure if the goal of attacking each model is to force the model to answer harmful questions or output toxic or biased outputs regardless of input. It should affect the judgment model as well, which is also not specified in the paper. The setup of the trojan attack is clear, not sure if the attack above shares the same setups and goals as the trojan attacks.
>
> > What are the setups of the jailbreaking attacks and what are the judgment models?
>
> We refer the reviewer to Sec. 3.1 for the detailed setup including the definition of jailbreaking attacks and the data used, as well as the judge models. As standard in the literature, the goal of a jailbreak is to make the target LLM comply with an unsafe request, which it would ordinarily, i.e. without jailbreak, refuse to follow (see examples in Fig.5 and Fig. 6).
>
> In Sec. 3.1, we also mention that we use GPT-4 as the judge model (a jailbreak is considered successful only if GPT-4 gives a 10/10 jailbreak score). Additionally,  we include results using the rule-based judge from [Zou et al. (2023)](https://arxiv.org/abs/2307.15043), as well as Llama-3-70B and Llama Guard 2 judges in Appendix B.6: our attacks are equally successful across different judges.
>
> ---
>
> > The proposed attack is still heavily emphasized on appending prefix or suffix tokens, which is not diverse or realistic enough.
>
> First, we want to emphasize that we use several techniques (prompt template, pre-filling, in-context prompt, self-transfer) which go beyond the simple suffix optimization as e.g. done by GCG. Second, the goal is to obtain successful jailbreaks, and for this it is not clear why one should refrain from using certain approaches (prefix/suffix) if these are effective. Finally, The fact that we achieve a 100% jailbreak success rate on deployed (even closed-source) LLMs like GPT-4o, Claude 3.5 and Llama-3 directly implies that our attacks are realistic.

---

> ### Author Response · Authors · 2024-11-24
> **Rebuttal (Part 2)**
>
> > What are the key technical challenges and contributions of the proposed method?
>
> We will try to emphasize this more clearly, but the main challenge of our paper was to find successful attacks that jailbreak leading **safety-aligned models** (i.e., models trained to withstand adversarial attacks) with 100% success rate **for the first time**. This was especially challenging since, e.g., models like Claude provide very restricted access: no gradients or even logprobs for iterative optimization.
>
> In the main part we primarily illustrate the positive results, i.e., attacks that are successful for a given model. However, Table 21 (in appendix) shows that several existing (from prior works) and sub-optimal approaches work only partially, e.g. `Prompt`/`Prompt + RS` for Llama-2-Chat-7B or `Transfer from GPT-4` for Claude-3-Opus. We believe this should give an impression about the challenges that we have encountered jailbreaking these models.
>
> ---
>
> > How to come up with the adversarial prompt templates? How much the proposed method will affect in performance of the template changes?
>
> As described in the paper, the main structure of the prompt template is kept the same across target LLMs, with only small adjustments for GPT-4o. Thus, we think that it can serve as a starting point for jailbreaking future models, to be refined on a case-by-case basis. For example, recent work has used our prompt template for jailbreaking LLM-based agents.
>
> ---
>
> > Can we improve the proposed attack by replacing random search with some more effective search, such as fuzzing or RL [1]?
>
> We expect it to be possible to improve the efficiency (not success rate, as it’s already at 100%) of the attacks  with more sophisticated optimization algorithms. However, one goal of our paper was to show that even simple methods, when properly combined, can provide very strong jailbreak attacks.
>
> ---
>
> > How to justify that the proposed method can test the model comprehensively and provide realistic attack prompts?
>
> The fact that we achieve 100% jailbreak success rate on deployed (even closed-source) LLMs like GPT-4o, Claude 3.5 and Llama-3 directly implies that our attacks are realistic. In fact, in this case, since the models we attack are the same used daily by millions of users, there is no disconnect between the research setup and the practical applications.
>
> Moreover, a 100% jailbreak success rate implies that the evaluation of existing LLMs is comprehensive (no better results can be achieved). It is however not guaranteed that the same techniques will be equally effective on future models, but they will nevertheless constitute a solid starting point for evaluating them, together with the guidelines we provide.

---

> ### Author Response · Authors · 2024-11-29
> **Discussion period ends soon**
>
> Dear Reviewer 2GYX,
>
> Thanks again for your detailed review. Since the discussion period is closing soon (December 2nd, Monday), we are wondering whether the revised version of our paper and our responses have addressed your concerns. If not, we would be happy to continue the discussion and provide further clarifications or explanations.

---

### Official Review · Reviewer_pkoY · 2024-11-02

**Soundness:** 3
**Presentation:** 3
**Contribution:** 2
**Rating:** 6
**Confidence:** 3

**Summary:**

The paper develops a prompt template that is modified with an adversarial suffix found through random search that jailbreaks all investigated models on nearly all investigated harmful requests. Furthermore, the random search is proposed to be initialized on adversarial suffixes found on easier-to-jailbreak requests. For Claude models where random search can't be performed because there is no access to the logprobs, a prefilling attack is proposed. Lastly, the paper provides a simple method to find triggers in models trained with backdoors.

**Strengths:**

* The paper is very well written and easy to follow.
* The proposed method outperforms the state-of-the-art in jailbreaking for a wide-range of current LLM models.
* The proposed method on finding triggers in backdoored models is simple and won a recent competition.
* The code is included in the submission.

**Weaknesses:**

1. The cornerstone of the method seems to be the static prompt template from Fig. 1, which is then slightly adapted with an adversarial suffix found through random search. Here, I wonder what is the actual scientific value and contributions to the scientific community. In particular, it was already known that safety-aligned LLMs can be jailbreaked and it's not surprising that with more engineering effort, the jailbreaking percentage can be further increased. In particular, there are no guidelines (or systematic method) presented on how the prompt template was created or could be adapted for other models. Such insights/guidelines for adaptive attack generation are e.g., provided by Tramèr et al. (2020) for image classification. That such insights will be given is mentioned in lines 93-98 in the main draft, however, I could not find such insights in the subsequent text.
2. The adaptivity of the attack is limited. This is related to the above point that it is based on a static prompt template reused for most models (except e.g., GPT-4o which has its own prompt template). While the random search of course adapts the method slightly to each model, an adaptive prompt design seems crucial (given e.g. 100% ASR of the prompt alone on GPT-3.5 Turbo, for which it was developed, and close to 0% on the other models).
3. The part on trojan detection seems rather disparate from the rest of the paper. In particular, in the related work, backdoor attacks for LLMs are not discussed and no single work on detecting triggers or on backdoor attacks in general is cited. (Backdooring is also never explained.)
4. I'm missing some experimental details and refer to the questions for this.

Minor weakness:
5. Slight overstatement. In line 82/83 and TL;DR 100% attack success rate is claimed on all leading safety-aligned LLMs. GPT-3.5 Turbo is defined as one of such leading safety-aligned LLMs. However, GPT-4 Turbo has an attack success rate of 96%.

**Questions:**

1. Self-transfer: I wonder how the authors concretely define a simple example? 100% ASR with pure RS? What if no sample has 100% ASR? Which of the "simple examples" do you choose to use to initialize the RS for the hard examples? Does it make a difference which ones you choose?
2. Prefilling: Can you provide details on how the prefilling option is exactly used?
3. From combinatorial optimization, it is known that initialization for random search is important (or for any local search algorithm). However, for me, it is not clear if the benefit is actually from the initialization point, or because the random search is given now "twice the budget". I.e., it can first search an adversarial suffix with N=10.000 and then optimize it for another prompt again with N=10.000.
4. What is the effectivity of the random search + self transfer without using the particular static prompt template? E.g., just providing the goal and adv_suffix? Connected, have you ablated parts of the prompt template?

Minor
* Link Appendix A.2 in Section 5, I was wondering what k you chose and by accident discovered you give the details there. I think, one could probably already include this information in the main text.
* Line 423: Abbreviation RLHF never introduced before.
* Figure 4 (Appendix B) is not color-blind friendly. I can't distinguish 15 from 25 tokens and 5 from 60 tokens.

Given the weaknesses and questions are appropriately addressed, I'm considering adapting my score.

**References**.
Tramèr et al. "On adaptive attacks to adversarial example defense", NeurIPS 2020

---

> ### Author Response · Authors · 2024-11-24
> **Rebuttal (Part 1)**
>
> We thank the reviewer for the detailed comments.
>
> ---
>
> > The cornerstone of the method seems to be the static prompt template from Fig. 1, which is then slightly adapted with an adversarial suffix found through random search. Here, I wonder what is the actual scientific value and contributions to the scientific community. In particular, it was already known that safety-aligned LLMs can be jailbreaked and it's not surprising that with more engineering effort, the jailbreaking percentage can be further increased.
>
> From a technical standpoint, we propose novel approaches for crafting jailbreaks attacks: 1) we show how to leverage random search for attacking LLMs which expose top-k logprobs, 2) we propose in-context jailbreaking and self-transfer, two novel techniques which contribute to successfully jailbreak GPT-4o, R2D2, and the Llama models, 3) we uncover how the pre-filling option in Claude models, which otherwise provide very restricted access, can be exploited for jailbreaks. These, together with the manually optimized prompt template, provide evaluation tools for future models. In particular, we expect new LLMs claiming better alignment (higher robustness to jailbreaks) will appear, and our work will be useful for testing them.
>
> Moreover, we managed to jailbreak leading **safety-aligned models** (i.e., models trained to withstand adversarial attacks) with 100% success rate **for the first time** and relatively simple approaches. Despite prior work showing the existence of jailbreaks, it could not achieve the same effectiveness: accurately measuring the robustness of current models is by itself relevant, as progress is not possible without first exactly quantifying the existing vulnerabilities.
>
> ---
>
> > In particular, there are no guidelines (or systematic method) presented on how the prompt template was created or could be adapted for other models. Such insights/guidelines for adaptive attack generation are e.g., provided by Tramèr et al. (2020) for image classification. That such insights will be given is mentioned in lines 93-98 in the main draft, however, I could not find such insights in the subsequent text.
>
> We would like to refer the reviewer to Sec. 6, where we provide recommendations for future evaluation as anticipated at the end of the introduction.
>
> As for giving precise guidelines on how to adapt the prompt template to new models, unfortunately this does not seem feasible in full generality, as different LLMs might require specific adjustments depending e.g. on the training data: this is typical of adaptive attacks, which have to be tailored for the target models. However, as shown in our paper and recent work which has used our prompt template for jailbreaking LLM-based agents, our template can be a good starting point for future evaluations.
>
> ---
>
> > The adaptivity of the attack is limited. This is related to the above point that it is based on a static prompt template reused for most models (except e.g., GPT-4o which has its own prompt template). While the random search of course adapts the method slightly to each model, an adaptive prompt design seems crucial (given e.g. 100% ASR of the prompt alone on GPT-3.5 Turbo, for which it was developed, and close to 0% on the other models).
>
> The adaptivity of our attacks is in the fact that we use different techniques depending on the characteristics of the target LLM: for example, Anthropic does not provide logprobs of Claude models, then we resorted on pre-filling rather than random search, or we use *in-context prompt* against R2D2 which was adversarially trained on single-turn suffix attacks. In general, we put forward the idea of adapting the attack to the specific weaknesses of each target model, without the need of a single method against every LLMs.
>
> ---
>
> > The part on trojan detection seems rather disparate from the rest of the paper. In particular, in the related work, backdoor attacks for LLMs are not discussed and no single work on detecting triggers or on backdoor attacks in general is cited. (Backdooring is also never explained.)
>
> In the trojan detection section, we show that the same approach used for jailbreaking LLMs (simple algorithm + adaptation to the specific goal) was successful in a different but still safety-related task. In fact, our solution relied on 1) narrowing down the set of candidate tokens via a task-specific observation (the adaptive part of the attack) and 2) optimizing the target score with random search (potentially the simplest zeroth-order algorithm). As for jailbreaking, this approach outperformed more sophisticated methods in the competition.
>
> We did not provide an extensive background on backdoor attacks since we do not propose a novel approach to such a problem, nor compare to existing methods, but just present a self-contained instance of the problem, i.e. the trojan detection competition. We would be happy to provide an extended background section in the appendix if it may be helpful to the reader.

---

> ### Author Response · Authors · 2024-11-24
> **Rebuttal (Part 2)**
>
> > Slight overstatement. In line 82/83 and TL;DR 100% attack success rate is claimed on all leading safety-aligned LLMs. GPT-3.5 Turbo is defined as one of such leading safety-aligned LLMs. However, GPT-4 Turbo has an attack success rate of 96%.
>
> Thanks for the comment: we considered GPT-4 Turbo as superseded by GPT-4o, while GPT-3.5 Turbo being from a different family of models (GPT-3.x). We will clarify this in the text.
>
> ---
>
> > Self-transfer: I wonder how the authors concretely define a simple example? 100% ASR with pure RS? What if no sample has 100% ASR? Which of the "simple examples" do you choose to use to initialize the RS for the hard examples? Does it make a difference which ones you choose?
>
> We refer as “simple example” to a request for which the attack succeeds with random search from the default initialization (i.e., exclamation marks separated by spaces). In practice this attack usually has a success rate >0%, so there is at least one adversarial suffix to transfer. If this was not the case, one could still transfer the string from a different model. If multiple simple examples are available, a random one is chosen for self-transfer.
>
> ---
>
> > Prefilling: Can you provide details on how the prefilling option is exactly used?
>
> For prefilling we force the LLM to start the generation from a string which repeats the unsafe request, e.g. “Sure, this is how to make a bomb” (see L226-230). We hypothesize this bypasses the safety guardrails since the model recognizes the unsafe request as part of its response rather than a question of the user, which is the format used during the safety-alignment.
>
> ---
>
> > From combinatorial optimization, it is known that initialization for random search is important (or for any local search algorithm). However, for me, it is not clear if the benefit is actually from the initialization point, or because the random search is given now "twice the budget". I.e., it can first search an adversarial suffix with N=10.000 and then optimize it for another prompt again with N=10.000.
>
> Fig. 2 shows that for Llama-2-7B the success rate without self-transfer (solid line) plateaus at around 40%, with minimal improvements from 5k to 10k iterations. However, when using self-transfers (dashed line), a few 100s iterations are sufficient to get the ASR to 100%. This clearly indicates that the advantage of self-transfer is much greater than simply using a higher number of iterations.
>
> ---
>
> > What is the effectivity of the random search + self transfer without using the particular static prompt template? E.g., just providing the goal and adv_suffix? Connected, have you ablated parts of the prompt template?
>
> Since GPT models provide only top-20 logprobs, the template is necessary to have the target token “Sure” among the 20 most likely to be generated, so that random search can be used for iterative refinement of the suffix. Even for open-weights models, for which *all* logprobs are always available, the template significantly improves the efficiency of the attacks.
>
> Just providing the goal and adv_suffix usually does not lead to a strong enough attack (e.g., the attack success rate on Llama-2-Chat-7B is around 10%). Moreover, providing the goal and optimizing the suffix is what GCG does, and our results show that random search + self-transfer with the prompt template consistently outperforms GCG.
>
> We have not computed a systematic ablation study on the rules of the prompt template. We will add this to the next version of the paper.
>
> ---
>
> > Link Appendix A.2 in Section 5, I was wondering what k you chose and by accident discovered you give the details there. I think, one could probably already include this information in the main text.
>
> We remark we report the value of $k$ for models 2, ..., 5 in L473 *in the main part*, we will add the same for model 1.
>
> ---
>
> > Line 423: Abbreviation RLHF never introduced before.
>
> Thanks for the suggestion, we will add it.
>
> ---
>
> > Figure 4 (Appendix B) is not color-blind friendly. I can't distinguish 15 from 25 tokens and 5 from 60 tokens.
>
> Thanks for the suggestion, we are happy to change the color scheme of the figure in the final version.

---

> ### Comment · Reviewer_pkoY · 2024-11-26
>
> I thank the authors for their responses. However, several of my concerns remain:
>
> 1. There are still no guidelines included on designing/adapting the crucial custom prompt template provided in the work for other models. The authors did refer to Sec. 6 in their answer, but Sec. 6 does not include such a discussion. I do not expect fully general and precise guidelines on how to adapt the prompt template to new models, but I do think discussing the approach taken and lessons learned on a higher level would be important such that the ICLR community can effectively build upon this work.
>
> 2. I'm, not convinced by the author's answers about the disparate trojan detection section in the paper. As this should be part of the paper and its contribution, I'm still expecting to include a discussion on the relevant background and related work in this field (I don't mind if given in the Appendix or main paper).
>
> 3. Many of the answers include "we will include this in the next version of the paper", but ICLR allows to submit an updated draft. I do think it is fine for some experimental results to not be provided full generality in the updated draft (e.g. ablating the prompt template), but I would expect to upload a revised draft including the discussions, clarifications, and experimental details as provided or promised by the authors.

---

> > ### Author Response · Authors · 2024-11-27
> > **Paper revision**
> >
> > We thank again the reviewer for the helpful comments.
> >
> > > I would expect to upload a revised draft including the discussions, clarifications, and experimental details as provided or promised by the authors.
> >
> > We have just uploaded a new revision that includes discussions, clarifications, and further details. We hope the new version addresses your concerns. If not, we are happy to discuss further!

---

> > > ### Comment · Reviewer_pkoY · 2024-11-28
> > >
> > > Thank you for your response. Can you point me to the sections in the revised paper where you address my remaining concerns 1 and 2?

---

> > > > ### Author Response · Authors · 2024-11-28
> > > > **Regarding concerns 1 and 2**
> > > >
> > > > **Concern 1**
> > > >
> > > > *Section 6* initially included a general discussion on our recommendations for adaptive attack evaluation of LLMs, similar in spirit as in the paper mentioned by the reviewer for $\ell_p$ robustness ([Tramer et al., 2020](https://arxiv.org/abs/2002.08347)):
> > > >
> > > > *"**Recommendations.** Thus, we believe it is important to use combinations of methods and identify unique vulnerabilities of target LLMs. First, the attacker should take advantage of the possibility to optimize the prompt template, which alone can achieve a high success rate (e.g., 100% on GPT-3.5). Second, standard techniques from the adversarial robustness literature can make the attack stronger, such as transferring an adversarial suffix or refining it iteratively via algorithms like random search, which may be preferred over gradient-based methods due to its simplicity and lower memory requirements. Finally, one can leverage LLM-specific vulnerabilities, for example by providing in-context examples or using the prefilling option. Importantly, in our case-study no single approach worked sufficiently well across all target LLMs, so it is crucial to test a variety of techniques, both static and adaptive."*
> > > >
> > > > In *Section 3.2* we initially wrote how we came up with the original prompt template:
> > > > *"We have optimized the set of rules one by one on the GPT-3.5 Turbo model to maximize the attack success rate and avoid the built-in safety guardrails."*
> > > >
> > > > But now we have expanded on this:
> > > > *"We have optimized the set of rules \textit{one by one} on GPT-3.5 Turbo to maximize the logprob of the first target token (e.g., token ``Sure''), which gives a very useful signal about which rules are effective against the built-in safety guardrails."*
> > > >
> > > > Also, we have added a clarification that the customized template for GPT-4o was obtained using a similar approach informed by the logprob of the target token:
> > > > *"Additionally, for GPT-4o, we customize the prompt template using the logprobs to guide manual search (i.e., similar to how we developed the main prompt template) and split it into a system and user parts, see Table 8 in Appendix B.1.*"
> > > >
> > > > ---
> > > >
> > > > **Concern 2**
> > > >
> > > > We have added this in *Section A.5: Extended Background and Related Work on Backdoor Attacks*. We have also updated the intro to *Section 5: Adaptive Attacks for Trojan Detection* to make the connection between the jailbreaking and trojan detection setting clearer.
> > > >
> > > > ---
> > > >
> > > > We hope these changes address your concerns. We thank again for all your feedback.

---

> > > > > ### Comment · Reviewer_pkoY · 2024-12-02
> > > > >
> > > > > I want to thank the authors for the updated draft and for pointing me to the respective sections.
> > > > >
> > > > > My Concern 2 is now well addressed. I also consider my Concern 3 (nearly fully) addressed. Here I would ask the authors to include the additional experimental details they provided in their response regarding self-transfer in Section 3 in the paper. Appendix A.3 now gives more details on the random search algorithm, which would be good to be linked in the main paper.
> > > > >
> > > > > I consider Concern 1 partially addressed. I do appreciate the additional details given on how the prompt template was designed. However, I do think that more expanded guidelines on how to make the attack adaptive also regarding the prompt template could still improve the paper.
> > > > >
> > > > > However, as the changes and responses by the authors could address all my concerns/weaknesses at least partially, I raise my score to 6. To conclude, for me, this is still a borderline paper, but I'm now leaning towards accept rather than reject.

---

### Official Review · Reviewer_82AN · 2024-11-02

**Soundness:** 3
**Presentation:** 2
**Contribution:** 3
**Rating:** 6
**Confidence:** 4

**Summary:**

The paper discusses methods for jailbreaking different frontier LLMs, achieving high success rates across many LLMs.

**Strengths:**

The results are fairly strong, but the insights from the paper are not so clear. The paper also has missing baselines and overclaim at points (see my review). To the authors: if you fix those, I would be happy to recommend acceptance.

- The results overall seem strong in terms of ASR, especially with the range of models tested. Good job.
- I like the self-transfer approach, this makes sense.

**Weaknesses:**

- I'm left uncertain what the main message or contribution of the paper is. Prefill? Log-prob random search? Adaptivity? I'd suggest trying to focus more on some key take aways, and linking the different sections of the paper. I also don't really understand how the backdoor / trojan section links to the rest of the paper. Though the approach in that section made sense.
- One weakness with the attack is that it would be extremely easy to detect with monitoring. E.g., the user making the request "don't start with "I can't assist with that"" is suspicious. I think this is fine, but please discuss this in the paper.
- There are many missing baselines, such as Many Shot Jailbreaking, AutoDAN. I think MSJ is likely to be an effective attack in particular, and I would expect it to get 100% success rate? Is there are reason you didn't include this?

**Questions:**

- *Pg 3: why are system prompt changes made for Claude? Are the results in Table 1 made with the modified system prompt? If so, Table 1 seems to be overclaiming.
- In table 1, I don't fully understand what the "prev" and "Ours" is? Because there are e.g., known jailbreaks on Claude Opus (Many Shot Jailbreaking, Pliny's Attacks). As a result, Table 1 seems to be overclaiming to me.
- How sensitivity are the results to the 10/10 GPT-4 Jailbreak Score?
- Do you search for universal adversarial suffixes? Or single ones?
- Line 220: the self-transfer approaches makes sense to me, I like it. The main question I have is how you are choosing / ranking the difficulty of different prompts?
- "Recent work showed the possibility of implanting backdoor attacks during the RLHF training
of LLMs by poisoning a small percentage of the preference data with a universal suffix." Please add citations.

---

> ### Author Response · Authors · 2024-11-24
> **Rebuttal (Part 1)**
>
> We thank the reviewer for the detailed comments. We discuss the weaknesses and questions below.
>
> ---
>
> ## Weaknesses
>
> > I'm left uncertain what the main message or contribution of the paper is. Prefill? Log-prob random search? Adaptivity? I'd suggest trying to focus more on some key take aways, and linking the different sections of the paper.
>
> The main takeaway of our paper is reflected in the title: **adaptive** attacks are crucial for accurate robustness evaluations of LLMs. All the attacks used in our work—including prefilling and log-prob random search—serve as examples of how model-specific adaptive attacks can look like. We will clarify this point better in the introduction and in the discussion section.
>
>
> > I also don't really understand how the backdoor / trojan section links to the rest of the paper. Though the approach in that section made sense.
>
> In the trojan detection section, we show that the same approach used for jailbreaking LLMs (simple random search algorithm + adaptation to the specific goal) was successful in a very related task that focuses on finding backdoored universal suffixes. In fact, our adaptive attack relied on 1) narrowing down the set of candidate tokens via a task-specific observation (the adaptive part of the attack) and 2) optimizing the target score with random search (potentially the simplest zeroth-order algorithm). As for jailbreaking, this approach outperformed more sophisticated methods in the competition.
>
>
> > One weakness with the attack is that it would be extremely easy to detect with monitoring. E.g., the user making the request "don't start with "I can't assist with that"" is suspicious. I think this is fine, but please discuss this in the paper.
>
> We agree that some of our attacks might be detectable: however, the fact that these are effective on currently deployed leading LLMs shows that such detection mechanisms are not currently in place. Moreover, we expect that in case more sophisticated defenses might appear, our approach would be a valuable starting point to bypass them. We will add discussion on this in the paper.
>
>
> > There are many missing baselines, such as Many Shot Jailbreaking, AutoDAN. I think MSJ is likely to be an effective attack in particular, and I would expect it to get 100% success rate? Is there are reason you didn't include this?
>
> We remark that we already include several baselines (when available) for each target LLM (see complete results in Table 21). Given the fast pace of the field and the cost of running some of the baselines, it is not feasible to compute the results of every existing method on every target model (some of these models have even appeared way after the attacks have been proposed). Moreover, since our attacks achieve a 100% success rate, they would outperform or match any other baseline.
>
> About the MSJ attack, the [original paper](https://openreview.net/pdf?id=cw5mgd71jW) shows in Fig. 19 that MSJ (128 shots) attacks achieve a success rate of 30-45% (depending on the version) against Claude 2.0, on the HarmBench behaviors similar to those of the baselines chosen in our paper. Conversely, our prompt + pre-filling attacks achieve 100% success rate (see Table 1 in our manuscript). Thus, it is not obvious that MSJ is a stronger attack than the baseline we report.
>
> We are nevertheless willing to add more baseline results to the paper, including MSJ.

---

> ### Author Response · Authors · 2024-11-24
> **Rebuttal (Part 2)**
>
> ## Questions
>
> > *Pg 3: why are system prompt changes made for Claude? Are the results in Table 1 made with the modified system prompt? If so, Table 1 seems to be overclaiming.
>
> Leveraging the system prompt is within our threat model, so we modify it to achieve higher attack success rates, as we write in page 8:
> *“The attack success rate of the transfer attack improves when the initial segment of the prompt (which corresponds to the set of rules to follow) is provided as the system prompt.”*
>
> The results in Table 1 for Claude were made with the modified system prompt, but we do not think it is overclaiming. We use publicly available API features of Claude, including the system prompt and prefilling option. Any potential adversary can leverage them, so we believe that allowing the use of these options should be part of any realistic threat model. We agree that we should be more explicit about it in Table 1, though. We will fix that.
>
>
> > In table 1, I don't fully understand what the "prev" and "Ours" is? Because there are e.g., known jailbreaks on Claude Opus (Many Shot Jailbreaking, Pliny's Attacks). As a result, Table 1 seems to be overclaiming to me.
>
> The [many-shot jailbreaking paper](https://openreview.net/pdf?id=cw5mgd71jW) does not show any results on Claude 3 Opus or any Claude 3 model.
>
> Pliny’s attacks are in general *not* competitive with state-of-the-art methods. E.g., a very recent paper [Endless Jailbreaks with Bijection Learning](https://arxiv.org/abs/2410.01294) (October 2, 2024) reports that the ASR of Pliny’s attacks is as follows on HarmBench:
>
> | Model             | Success rate of Pliny’s attacks   |
> |--------------------|---------|
> | Claude 3.5 Sonnet | 50.0%   |
> | Claude 3 Haiku    | 15.9%   |
> | Claude 3 Opus     | 65.9%   |
>
> We will, of course, be happy to add these results to Table 1.
>
>
>
> > How sensitivity are the results to the 10/10 GPT-4 Jailbreak Score?
>
> Most of the time, GPT-4 as a judge outputs either a 1/10 or 10/10 score and rarely anything in between. We use it with temperature 0, so GPT-4 is almost deterministic, except for minor variations due to inference time randomness. Moreover, in Table 20 of the appendix, we include an experiment comparing the attack success rates of different judges. These results suggest that the evaluation is not highly sensitive to the choice of judge.
>
> | Experiment               | GPT-4  | Rule-based | Llama-3-70B | Llama Guard 2 |
> |--------------------------|--------|------------|-------------|---------------|
> | Vicuna-13B              | 100%   | 96%        | 100%        | 96%           |
> | Mistral-7B              | 100%   | 98%        | 100%        | 98%           |
> | Phi-3-Mini-128k         | 100%   | 98%        | 100%        | 86%           |
> | Gemma-7B                | 100%   | 98%        | 100%        | 90%           |
> | Llama-2-Chat-7B         | 100%   | 90%        | 100%        | 88%           |
> | Llama-2-Chat-13B        | 100%   | 96%        | 100%        | 92%           |
> | Llama-2-Chat-70B        | 100%   | 98%        | 100%        | 90%           |
> | Llama-3-Instruct-8B     | 100%   | 98%        | 100%        | 90%           |
> | R2D2                    | 100%   | 98%        | 100%        | 96%           |
> | GPT-3.5 Turbo           | 100%   | 90%        | 100%        | 94%           |
>
>
>
> > Do you search for universal adversarial suffixes? Or single ones?
>
> We always search for *request-specific* and *model-specific* suffixes, but some of them happen to be transferable (e.g., a suffix transferred from GPT-4 to Claude 3.5 Sonnet leads to 96% ASR when combined with the prompt template).
>
>
> > Line 220: the self-transfer approaches makes sense to me, I like it. The main question I have is how you are choosing / ranking the difficulty of different prompts?
>
> For this, we track the logprob of the target token that we optimize for. The hardest examples correspond to the lowest logprobs. We will clarify this in the paper.
>
>
> > "Recent work showed the possibility of implanting backdoor attacks during the RLHF training of LLMs by poisoning a small percentage of the preference data with a universal suffix." Please add citations.
>
> We did not add the citation here since the authors of this paper also organized the trojan detection competition in which we participated. Thus, including a direct reference could potentially reveal our identities. Instead, we will clarify the overall setting better.
>
> ---
>
> > if you fix those, I would be happy to recommend acceptance.
>
> We hope our response addresses your concerns. We are happy to provide more clarifications if needed.

---

> > ### Author Response · Authors · 2024-11-27
> > **Paper revision**
> >
> > Thanks a lot for your detailed feedback. We have just uploaded a revised version of our paper that incorporates the points above.
> >
> > > To the authors: if you fix those, I would be happy to recommend acceptance.
> >
> > We hope our changes address your concerns.

---

> > > ### Comment · Reviewer_82AN · 2024-12-02
> > >
> > > Thanks, I've updated my score

---

### Official Review · Reviewer_LQ5H · 2024-11-04

**Soundness:** 3
**Presentation:** 3
**Contribution:** 2
**Rating:** 6
**Confidence:** 3

**Summary:**

This paper proposes an adaptive jailbreaking attack, which aims at attacking safety-aligned language models (LLMs), demonstrating that even the latest models are vulnerable to these attacks. The authors use targeted prompts and random search methods to bypass safety measures in models, achieving a 100% attack success rate across multiple models with additional enhanced methods. The experimental results include a range of models and APIs, showing that approaches like self-transfer, prefilling, and targeted adversarial suffixes effectively achieve jailbreaks across various LLMs. Additionally, the paper reports success in trojan detection by applying similar techniques.

**Strengths:**

1. This paper focuses on black-box models. The proposed method can have a very high attack success rate on close-source LLMs such as GPT and Claude.

2. The analysis of self-transfer is insightful.

3. Beyond jailbreaking, the paper successfully applies its adaptive techniques to trojan detection.

**Weaknesses:**

1. Though the proposed method is designed for black-box models, attacking black-box models such as GPT4o highly relies on the powerful initial prompt. And in the paper, the authors use a custom prompt for GPT4o, which restricts the usage of the proposed method.

2. The search space for the proposed method is very large, how to effectively search the replaced token is a problem that this paper does not discuss (for jailbreaking).

3. The attack strategies are tailored to individual models, which limits their generalizability and may require frequent updates as models evolve.

**Questions:**

1. How do the authors guarantee the effectiveness of random search?

---

> ### Author Response · Authors · 2024-11-24
> **Rebuttal**
>
> We thank the reviewer for the detailed comments. We discuss the weaknesses and questions below.
>
> ---
>
> > Though the proposed method is designed for black-box models, attacking black-box models such as GPT4o highly relies on the powerful initial prompt. And in the paper, the authors use a custom prompt for GPT4o, which restricts the usage of the proposed method.
>
> We do not view this as a weakness of our work. On the contrary, this case strongly supports our main message about the importance of adaptive attacks: if an attack method fails on a particular model (possibly because the model has been trained on that specific attack) it does not imply the model is truly robust. Instead, the most straightforward approach is to adapt the attack. For example, for GPT-4o, a slight modification of the prompt template was enough to completely jailbreak it.
>
>
> ---
>
>
> > The search space for the proposed method is very large, how to effectively search the replaced token is a problem that this paper does not discuss (for jailbreaking).
>
> We did not discuss this for jailbreaking since we have not found any scheme that would work better than naive sampling from the whole token vocabulary at each iteration of random search. We tried to restrict the search space only to tokens that contain latin characters, for example, but this led to worse performance. For trojan detection, however, we have found a very effective scheme (as described in Section 5) that significantly restricts the search space and leads to substantial gains in terms of final scores.
>
> ---
>
>
> > The attack strategies are tailored to individual models, which limits their generalizability and may require frequent updates as models evolve.
>
> Again, we believe that this is not a weakness but instead the most important takeaway of our paper: if one wants to accurately evaluate the robustness of a model, one has to use adaptive attacks which are, by definition, tailored to individual models. Using only standardized attacks very often leads to a false sense of robustness. This principle also applies to the attacks introduced in our paper: using them alone is insufficient.  One has to explore new attacks and discover unique weak points of a target model.
>
> ---
>
>
> > How do the authors guarantee the effectiveness of random search?
>
> Our empirical results provide clear evidence of the effectiveness of random search, particularly when used in combination with a strong prompt template. However, we do not have any formal guarantees that random search will be effective in every scenario and for every model. We are also not aware of any work that provides such guarantees against jailbreaking attacks on LLMs.
>
> ---
>
> We thank the reviewer again and hope the original score can be reconsidered in light of our clarifications.

---

> > ### Comment · Reviewer_LQ5H · 2024-11-25
> >
> > Thanks to the authors for the detailed reply. However, I still have some questions regarding the main purpose of the paper as well as the effectiveness of the random search.
> >
> > - If the main message of the paper is about the importance of adaptive attacks—specifically, that if an attack method fails on a particular model (possibly because the model has been trained on that specific attack), it does not imply the model is truly robust—then there is a fundamental issue. The paper does not provide a formal definition of adaptive attacks, nor does it establish any guarantee that such attacks exist for all cases. This makes robustness evaluation inherently problematic. For example, under this definition, we could never claim that a model is robust, as it is impossible to exhaustively test against all possible adaptive attacks (e.g., all customized prompts). This limitation highlights the need for a general methodology to efficiently identify adaptive attacks for different models rather than relying solely on ad hoc customizations.
> >
> > - I still don’t quite understand the effectiveness of random search. For example, the vocabulary size of Gemma is 256,000 tokens. If random search is applied without any improvements, the authors would need to randomly sample one token out of 256,000 for each position, and this process must be repeated 25 times for a suffix length of 25. Yet, the paper claims that only 1,000 trials suffice to solve the problem, which seems counterintuitive. The authors should include more ablation studies or detailed statistics to substantiate this claim and address my concern.
> >
> > Given the remaining concerns, I will maintain my current score.

---

> > > ### Author Response · Authors · 2024-11-25
> > > **Thank you for the follow-up comment**
> > >
> > > Thank you for the follow-up comment!
> > >
> > > ---
> > >
> > > > The paper does not provide a formal definition of adaptive attacks
> > >
> > > We mention what we mean by adaptive attacks directly in the introduction (Lines 050 - 052) and at the beginning of Section 3.2 (Lines 165-167):
> > > *"... adaptive attacks, which we define as attacks that are specifically designed to target a given defense [(Tramer et al., 2020)](https://arxiv.org/abs/2002.08347)."*
> > >
> > > Note that we simply use an established definition from the literature on adversarial robustness.
> > >
> > >
> > > > nor does it establish any guarantee that such attacks exist for all cases
> > >
> > > We are not sure how one can derive such guarantees. If there is any work in this direction, we would be happy to learn about it.
> > >
> > >
> > > > For example, under this definition, we could never claim that a model is robust, as it is impossible to exhaustively test against all possible adaptive attacks (e.g., all customized prompts).
> > >
> > > That's correct, we could never claim that a model is robust just by evaluating it against a set of attacks. The only way to escape this cat-and-mouse game is to derive formal provable guarantees as has been done for Lp-robustness ([Raghunathan et al. (2018)](https://arxiv.org/abs/1801.09344)). However, we are not aware of any non-vacuous works in this directions that would be able to certify robustness of frontier LLMs to jailbreak attacks. Thus, we believe that performing adaptive attacks (as opposed to evaluation with a fixed set of attacks) is the best way forward.
> > >
> > > ---
> > >
> > > > If random search is applied without any improvements, the authors would need to randomly sample one token out of 256,000 for each position, and this process must be repeated 25 times for a suffix length of 25. Yet, the paper claims that only 1,000 trials suffice to solve the problem, which seems counterintuitive.
> > >
> > > We would like to clarify how exactly we apply random search. To optimize the adversarial suffixes, we use an algorithm first introduced in [Rastrigin et al. (1963)](https://archive.org/details/sim_automation-and-remote-control_1963-11_24_11/page/n1/mode/2up?view=theater) but we adapt it for the LLM setting. Our algorithm proceeds as follows:
> > > - First, we append a suffix of a specified length to the original request (we use 25 tokens and provide an ablation study for this choice in Figure 4 in the appendix).
> > > - In each iteration (out of 10'000), we modify **a few contiguous tokens** at a random position in the suffix. Each substitution token is sampled randomly from the vocabulary (which can be indeed as large as 256'000). The number of contiguous tokens at each iteration is selected according to a pre-defined schedule. This parameter plays a role similar to the learning rate in continuous optimization.
> > > - Then, in each iteration, we accept the modified tokens **if they help to increase the log-probability of a target token** at the first position of the response. We use the token ``Sure'' as the target token, unless mentioned otherwise, that leads the model to comply with a harmful request.
> > >
> > > Thus, we do not necessarily sample only one token at each iteration. Moreover, the suffix length (such as 25) does not determine the total number of iteration. We will include a precise algorithm based on the description above in the revised version of the paper.

---

> ### Comment · Reviewer_LQ5H · 2024-11-25
> **Thanks for the authors' reply**
>
> Thanks for the quick reply from the authors. Let me clarify my concerns more clearly:
>
> 1. The first concern actually comes from the paper's main purpose. If the paper's main purpose is what the authors said in the first reply
> ```Again, we believe that this is not a weakness but instead the most important takeaway of our paper: if one wants to accurately evaluate the robustness of a model, one has to use adaptive attacks which are, by definition, tailored to individual models. ```
>
> Then, the authors should provide a more formal definition of adaptive attack so that we can know which adaptive attack should we use to evaluate the robustness. For example, based on the current definition, GCG is also an adaptive attack because the gradient computed during GCG is adapted to different questions and different models so that the chosen adversarial suffixes are also adapted. However, given the results in the paper, GCG is not good for measuring the real robustness given the performance gap between GCG and the proposed method.
>
> Besides, if we really need to care about how to **accurately evaluate the robustness of a model**, the contradictory arises since the authors claim that
>
> ``` That's correct, we could never claim that a model is robust just by evaluating it against a set of attacks.  The only way to escape this cat-and-mouse game is to derive formal provable guarantees as has been done for Lp-robustness (Raghunathan et al. (2018)). ``` ,
>
> which means we cannot evaluate robustness accurately without certified robustness. Then how could we say:
>
> ```if one wants to accurately evaluate the robustness of a model, one has to use adaptive attacks which are, by definition, tailored to individual models. ```?
>
> On the other hand, if the paper's main purpose is introducing a new adaptive attack method,  then we should care about the generalizability of this adaptive attack method. Using the strong custom prompt is also one procedure of this adaptive attack and thus I ask **how could we find such a strong custom initial prompt for every model**.
>
> 2. The explanation of steps in random search does not address my concern. For example, if we are optimizing 3 continuous tokens, then the sampling space has a total number of actions around **1.5E16** ($256000 \times 256000 \times 256000$) since each substitution token is sampled randomly from the vocabulary with a size 256000. Then **10000** trial is so little compared with **1.5E16**. Only using 10000 (which is actually stated as 1000 for Gemma-7B in the paper) trial could get a very good result seems counterintuitive to me. The conclusion behind this phenomenon is that it is highly likely ($>99.999$%) that a random token will increase the logprob of 'sure'. What I want to see is whether this phenomenon is actually true or if the authors have another explanation for why random search will work.

---

> > ### Author Response · Authors · 2024-11-25
> > **Thank you for the quick reply and constructive discussion.**
> >
> > Thank you for the quick reply and constructive discussion.
> >
> > ---
> >
> > > For example, based on the current definition, GCG is also an adaptive attack because the gradient computed during GCG is adapted to different questions and different models so that the chosen adversarial suffixes are also adapted.
> >
> > GCG, as any fixed attack, is not adaptive in the sense used in the literature on adversarial robustness (i.e., as defined in [On Evaluating Adversarial Robustness](https://arxiv.org/abs/1902.06705)). The adaptivity that we refer to concerns selection of the **attack method**. Our main point is that using any fixed attack, like GCG, is prone to robustness overestimation. Thus, it is important to change the attack method depending on unique vulnerabilities of a given model. A prominent example here is the Claude family: their models are quite robust to conventional attacks, but they break completely when we leverage the prefilling option available via the API. This is an adaptive attack because it's a unique vulnerability of the Claude API (which is, for example, not available for GPT models).
> >
> > ---
> >
> > > Besides, if we really need to care about how to accurately evaluate the robustness of a model, the contradictory arises since the authors claim that
> > `That's correct, we could never claim that a model is robust just by evaluating it against a set of attacks. The only way to escape this cat-and-mouse game is to derive formal provable guarantees as has been done for Lp-robustness (Raghunathan et al. (2018)).`
> > > which means we cannot evaluate robustness accurately without certified robustness.
> >
> > There is one exception to this: if one achieves 100% attack success rate, then one has a proof (i.e., the set of prompts and outputs) that adversarial robustness has been evaluated accurately. Any number less than 100% can always be questioned without having provable robustness guarantees. This has been a recurring theme of the research in adversarial robustness—e.g., as highlighted in [Obfuscated Gradients Give a False Sense of Security: Circumventing Defenses to Adversarial Examples](https://arxiv.org/abs/1802.00420).
> >
> > ---
> >
> > > I ask how could we find such a strong custom initial prompt for every model.
> >
> > It is probably impossible to formulate comprehensive guidelines for this. This is precisely the reason why human red-teaming is still very important: to find model vulnerabilities, we constantly need new creative approaches.
> >
> > ---
> >
> > > Then 10000 trial is so little compared with 1.5E16.
> >
> > If we get a useful signal on each iteration of an optimization algorithm, the whole procedure tends to succeed. This is the same principle as in optimization of deep networks with billion parameters: although the space of possibilities is infinite, gradient-based method in practice do succeed. Note that we still have no formal guarantees for deep learning optimization, i.e., strictly speaking, we don't know when and why gradient descent succeeds. The same applies for random search: we can only state that it works empirically on the models we have evaluated.

---

> > > ### Author Response · Authors · 2024-11-27
> > > **Paper revision**
> > >
> > > To further address your concerns, we have uploaded a revised version of the paper. In particular, following your suggestions, we have added a further discussion on what we precisely mean by adaptive attacks in **Section A.2: Definition of Adaptive Attacks**. We have also included there a formal definition of adaptive attacks.
> > >
> > > We have also added a formal description of our algorithm in **Section A.3: Random Search Algorithm** and multiple other changes requested by other reviewers.
> > >
> > > We hope this revision clarifies the concerns we have been discussing!

---

> > > > ### Comment · Reviewer_LQ5H · 2024-11-28
> > > > **Thanks for the author's effort**
> > > >
> > > > Thank you for the follow-up discussion and clarification. While I appreciate the authors' efforts in addressing my concerns, I remain worried about the effectiveness and explanation of random search. In the context of deep learning optimization, gradient-based methods provide a clear and interpretable mechanism for navigating the parameter space, as the gradient serves as a directional signal to guide the search. Although there may not be formal guarantees for the global success of gradient-based optimization, each step of gradient descent is grounded in well-understood principles and offers an intuitive pathway toward improvement.
> > > >
> > > > By contrast, random search lacks such directional guidance, which makes its success less immediately intuitive, especially in the context of extremely large search spaces. However, I acknowledge the empirical effectiveness demonstrated by the proposed methods and the diverse attack strategies presented in the paper. Given these contributions, I would like to raise my score to 6.

---

### Official Review · Reviewer_v3C3 · 2024-11-04

**Soundness:** 2
**Presentation:** 2
**Contribution:** 3
**Rating:** 8
**Confidence:** 4

**Summary:**

This paper presents results for adaptive attacks on a variety of models. They demonstrate that it is possible to circumvent existing defenses by using attacks targeted to the model/defense. Their attack uses a random search based attack to generate an adversarial suffix, in addition to a prompt customized to the model being attacked. They demonstrate that their attack can successfully attack GPT-4o and other closed source models, in addition to a range of open source models.

**Strengths:**

1. This paper demonstrates that existing defenses are quite easy to circumvent using adaptive attacks. The combination of prompt engineering and adversarial suffix search is a good demonstration of how different techniques may be combined to effectively attack models.

2. Especially as research on defenses and attacks progress, adaptive attacks become more realistic possibilities. This is a good demonstration of their effectiveness.

3. Experiments are performed on a wide range of models, both open and closed source.

4. The efficacy of the random-search based attack is important, and demonstrates that closed-source models are more vulnerable than previously thought.

**Weaknesses:**

Most weaknesses in this paper have to do with organization or comparison to prior results. However, they are significant enough that I am not comfortable accepting the paper without these points resolved:

## 1. Comparison to prior work

My main concern is that the evaluation comparisons are very inconsistent. Whereas this work presents their own evaluations using a split of 50 samples from AdvBench, they present comparisons to prior work using different data and evaluation methods. While there are notes on this in the text, this is not a fair comparison of results. While the results presented are convincing, the lack of adequate comparison makes it very hard to situate this work in existing literature and will make it very hard to build upon. The prior attacks are also not clearly referenced in the main summary in Table 1, making it impossible to see what is being compared against. Comparisons between this work and prior work needs to be standardized.

## 2. Organization
The organization of the paper is somewhat hard to follow, particularly when it comes to Section 5. It was not entirely clear to me how this section fits with other results in the paper, and I feel that it could be better contextualized.

## 3. Data documentation
Documentation on data used for training and testing is vague. For Table 1, it says results are reported on a subset of 50 samples from AdvBench, however it is not clear what this subset is, or whether this is the set of 50 that is used to train the attack.

**Questions:**

1. What is considered a significant number of false positives by the LLM judge?

2. Is the training and test data distinct for this attack? It is not clear from the text, which mentions a set of 50 samples from AdvBench for both.

---

> ### Author Response · Authors · 2024-11-24
> **Rebuttal**
>
> We thank the reviewer for the detailed comments. We address their concerns below together with new results.
>
> ---
>
> ### Comparison to prior work
>
> We agree that the comparison to prior work could have been more direct. To address this concern, here we present a fair comparison with [GCG](https://arxiv.org/abs/2307.15043), [PAIR](https://arxiv.org/abs/2310.08419), and a popular template attack (“Always Intelligent and Machiavellian” (AIM) from http://jailbreakchat.com/) on 100 JBB-Behaviors from JailbreakBench using a different semantic judge (Llama-3 70B instead of GPT-4 with a different prompt). The numbers below are attack success rates for four different models:
>
> | Attack           | Vicuna 13B | Llama-2-Chat-7B | GPT-3.5 | GPT-4 Turbo |
> |-------------------|--------|---------|---------|-------|
> | GCG              | 80%    | 3%      | 47%     | 4%    |
> | PAIR             | 69%    | 0%      | 71%     | 34%   |
> | Jailbreak-Chat          | 90%    | 0%       | 0%      | 0%    |
> | Prompt Template + Random Search + Self-transfer  | 89%    | 90%     | 93%     | 78%   |
>
> We would like to highlight the particularly large gap on Llama-2-Chat-7B: 90% ASR with our method vs. 3% ASR of plain GCG. Also, we note that the Llama-3 70B judge is in general *stricter* than the GPT-4 judge used as a stopping criterion for the attack, which explains a general drop of ASRs from 100% to the 80%-90% range. However, we expect that with more random restarts and by using the target judge model as a stopping criterion, the ASR of our attacks will go up to 100%. We will add and discuss these results in the paper.
>
>
> > The prior attacks are also not clearly referenced in the main summary in Table 1, making it impossible to see what is being compared against.
>
> We omitted the attack names since we didn’t have enough horizontal space in the table. We will clarify that the names and citations of the methods are available in the subsequent tables (i.e., Tables 2, 3, 4).
>
> ---
>
> ### Organization
>
> Section 5 describes the task of finding universal trojan strings—which are suffixes appended to inputs—in poisoned models, which is an almost identical setting to standard jailbreaking. We will explain this setting better and connect it better with Section 4.
>
>
> ---
>
> ### Data documentation
>
> Actually, there is no distinction between training and testing, as we *do not* optimize for universal attacks in our paper. All our adversarial suffixes are optimized to be *request-specific* and *model-specific*. At the same time, some of our suffixes came out to be universal *“by accident”* (i.e., without explicitly optimizing for it), as our transfer results from GPT-4 to Claude 3.5 Sonnet suggest (96% ASR). We will make this fact clearer in the paper.
>
> ---
>
> ### Questions
>
> > What is considered a significant number of false positives by the LLM judge?
>
> We estimate it at around 20% for Claude 2.1. All other models have a negligible amount of false positives (<5%).
>
> > Is the training and test data distinct for this attack? It is not clear from the text, which mentions a set of 50 samples from AdvBench for both.
>
> We will make it clearer in the paper that there is no distinction between the training and test data, as all our suffixes are request-specific and model-specific. Thus, there are no held-out requests or models.
>
> ---
>
> We thank the reviewer again and hope that we have addressed their main concerns.

---

> > ### Comment · Reviewer_v3C3 · 2024-12-02
> >
> > Thank you for clarifying these points. This addresses my concerns and I'm comfortable raising my score.

---

### Official Review · Reviewer_dr4H · 2024-11-12

**Soundness:** 3
**Presentation:** 3
**Contribution:** 3
**Rating:** 6
**Confidence:** 3

**Summary:**

In their study, the authors explore the vulnerability of safety-aligned Large Language Models (LLMs) to adaptive jailbreaking attacks, demonstrating that even models designed for robustness against harmful prompts can be compromised. They employ a methodology based on modifying adversarial prompts and optimizing suffixes via random search to maximize target log probabilities, achieving a 100% success rate across several LLMs including Vicuna-13B, Mistral-7B, and different versions of GPT and Claude. This paper underscores the adaptability required in attack strategies, tailored to exploit specific model vulnerabilities and interfaces, and extends their findings to the detection of trojan strings in poisoned models, highlighting significant security concerns for real-world applications of LLMs.

**Strengths:**

- This paper raises an interesting perspective on the safety vulnerabilities in LLMs, highlighting the critical need for ongoing advancements in AI security to effectively address these emerging threats.

- This paper is commendable for its extensive experimental coverage, which spans a wide array of popular LLMs, providing a comprehensive understanding of the techniques’ effectiveness across different model architectures and configurations.

**Weaknesses:**

- The paper heavily utilizes custom prompt templates that instruct the LLM to bypass safety mechanisms by avoiding certain words and phrases, which appears to significantly contribute to the success of the attacks. However, the roles and effectiveness of random search and self-transfer techniques are not thoroughly discussed beyond demonstrating an increase in attack success rates. A deeper exploration of how these methods specifically enhance the attacks could provide a clearer picture of their contribution and potential for generalization across different models.

 - While the paper provides interesting insights into exploiting LLM vulnerabilities through prompt engineering, it seems to lack depth in novel algorithmic development or thorough analytical rigor.

**Questions:**

- Can the authors provide more detailed explanations of how the adversarial suffixes were optimized? What specific characteristics of these suffixes make them effective in bypassing the safety mechanisms of LLMs?

- The study employs a variety of custom prompts combined with different techniques across several LLMs, leading to diverse outcomes. Can the authors provide a deeper analysis or discussion on how different prompt configurations influenced the effectiveness of attacks across the models tested?

---

> ### Author Response · Authors · 2024-11-24
> **Rebuttal (Part 1)**
>
> We thank the reviewer for the detailed comments. We comment on the weaknesses and questions below.
>
>
> ---
>
>
> ## Weaknesses
>
> > However, the roles and effectiveness of random search and self-transfer techniques are not thoroughly discussed beyond demonstrating an increase in attack success rates. A deeper exploration of how these methods [prompt templates, random search, self-transfer] specifically enhance the attacks could provide a clearer picture of their contribution and potential for generalization across different models.
>
> We would like to clarify that the roles and effectiveness of random search and self-transfer are presented in Figure 2 for multiple models (Llama-3-Instruct-8B, Llama-2-Chat-7B, and Gemma-7B). Starting from a good initialization via self-transfer is *crucial* not only for a high attack success rate but also for query efficiency.
>
> Moreover, in Tables 2 and 3, we provide side-by-side results for various settings: `Prompt`, `Prompt + Random Search`, and `Prompt + Random Search + Self-Transfer`. E.g., for Llama-2-Chat-7B, the attack success rate is 0% for `Prompt`, 50% for `Prompt + Random Search`, and 100% for `Prompt + Random Search + Self-Transfer`. In addition, extended results are available in Table 21 (the last table of the appendix). These results clearly demonstrate the importance of all three components of the attack. We hope these explanations clarify the role of random search and self-transfer.
>
>
> > While the paper provides interesting insights into exploiting LLM vulnerabilities through prompt engineering, it seems to lack depth in novel algorithmic development or thorough analytical rigor.
>
> We made multiple important algorithmic developments:
> - We showed how to successfully adapt the random search algorithm for the jailbreaking setting of LLMs and for trojan detection. We note that implementation specific—such as restricting the token search space for the trojan detection task—are key to make it work. In addition, our results imply that the gradient information for attacks like GCG is not necessary to find effective and transferable adversarial suffixes.
> - We introduced the prefilling attack as a strong, optimization-free method that successfully jailbreaks models such as Claude, which are very difficult to reliably jailbreak with existing methods.
> - We came up with a powerful manually written template that has been reused many times in subsequent works.
> - We introduced the “self-transfer” technique which is key for query efficiency and high attack success rates.
>
> Only through a combination of **all** these techniques we were able to achieve 100% attack success rate on leading LLMs, including very recent ones, such as GPT-4o and Claude 3.5 Sonnet. Moreover, subsequent works have successfully reused our jailbreak template, prefilling attack, and pre-computed random search suffixes, including in the LLM agent setting. We believe this confirms that our methodological contribution is substantial.
>
> –--
>
>
> ## Questions
>
> > Can the authors provide more detailed explanations of how the adversarial suffixes were optimized?
>
> To optimize the adversarial suffixes, we use a simple random search (RS) algorithm first introduced in [Rastrigin et al. (1963)](https://archive.org/details/sim_automation-and-remote-control_1963-11_24_11/page/n1/mode/2up?view=theater) but we adapt it for the LLM setting. Our algorithm proceeds as follows:
> - First, we append a suffix of a specified length to the original request (we use 25 tokens and provide an ablation study for this choice in Figure 4 in the appendix).
> - In each iteration, we modify a few contiguous tokens at a random position in the suffix. The number of contiguous tokens at each iteration is selected according to a pre-defined schedule. This parameter plays a role similar to the learning rate in continuous optimization.
> - Then, in each iteration, we accept the modified tokens if they help to increase the log-probability of a target token at the first position of the response. We use the token ``Sure'' as the target token, unless mentioned otherwise, that leads the model to comply with a harmful request.
>
> We use up to 10,000 iterations and up to 10 random restarts of the whole procedure, although in most cases a single restart suffices.
>
> We will clarify our method in the paper by providing a clear algorithm in the appendix.

---

> > ### Comment · Reviewer_dr4H · 2024-11-26
> >
> > Thanks for addressing my comments. I will raise the score.

---

> ### Author Response · Authors · 2024-11-24
> **Rebuttal (Part 2)**
>
> > What specific characteristics of these suffixes make them effective in bypassing the safety mechanisms of LLMs?
>
> Here is an example of a transferable suffix optimized on GPT-4o that leads to a 96% attack success rate on Claude 3.5 Sonnet (when combined with our prompt template):
> ```
> <bos>business Gl headed konk;?>
>  usual_TOOcr	video Section wool Funktion double運行rač calculations comfortable的Defs cambiando**forder.AllArgsConstructoryssey.tem/dockerriamemo/INFO Damon,.(ExpectedConditionser_Entity stylKuva GENERATION Mudquido quantumentai(isinstance_packet的功能
> ```
>
> This suffix does not seem to contain clear instructions for the LLM. Our best guess is that the suffix helps distract the safety mechanism of the LLM from paying attention to harmful content. In this way, it can potentially act as an out-of-distribution example compared to the examples encountered during the model’s safety training/fine-tuning.
>
>
> > Can the authors provide a deeper analysis or discussion on how different prompt configurations influenced the effectiveness of attacks across the models tested?
>
> To clarify, we used only two substantially different prompts: the main template shown in Table 1 and the in-context template shown in Table 6. We used the main template for almost all models except R2D2, for which it was ineffective (12% attack success rate with random search). Meanwhile, the in-context template worked with a 100% attack success rate on R2D2 when combined with random search. We also compared the two templates on Llama-2-Chat-7B (see Table 21 in the appendix): the main template led to a 100% ASR, while the in-context template led to a 76% ASR.
>
> ---
>
>
> We thank the reviewer again and hope that we have addressed their concerns. We also hope that the original score can be reconsidered in light of our clarifications.

---

### Author Response · Authors · 2024-11-27
**Our revised PDF is uploaded now**

Dear Reviewers,

We thank you for your feedback, which has been helpful in improving and expanding the paper. We have taken your comments very seriously and uploaded a revised version.

We would like to note that we received 7 reviews, significantly above the average, which required more time to provide detailed responses. Similarly, revising the paper and adding 4 pages of new content to address your concerns also took considerable effort.

In addition to updates to the main paper, we have added a new Section A in the appendix to address larger concerns raised during the discussion phase. For clarity, we have grouped these updates into a single section for now, but we plan to integrate them throughout the appendix in the next version.

Thank you again for your feedback and responsiveness. We greatly value your input and look forward to further discussions!

Best regards,
The Authors

---

### Meta-Review · Area_Chair_S2of · 2024-12-17

**Metareview:**

The reviewers largely agreed that this paper should be accepted.  Even though I find the technical novelty in this paper to be highly limited, most of this wisdom only exists on Twitter at the moment, so I think a paper on this content is worth publishing.  Therefore, I recommend acceptance.  The reviewers brought up a number of concerns, which the authors have largely addressed.  For example, reviewers requested a deeper exploration of the techniques used, but I think the paper demonstrates the effectiveness of their techniques, including numerous ablations, and I think a mechanistic study can be left for the future.  The authors have also included additional comparison to competing methods and have promised to add more in the camera ready.  Reviewers also mentioned limitations of using custom prompts, but I think it is valuable to see just how effective strong attacks can be.  Overall, both I and the reviewers view the paper favorably despite any concerns.

**Additional Comments On Reviewer Discussion:**

The authors engaged strongly with the reviewers during the rebuttal period and have addressed most negative feedback.

---

### Decision · Program_Chairs · 2025-01-22

Accept (Poster)